| Editor's Pick | Ecology | Research Article

# Genomic fingerprints of the world's soil ecosystems

Emily B. Graham,[1,2] Vanessa A. Garayburu-Caruso,[1] Ruonan Wu,[1] Jianqiu Zheng,[1] Ryan McClure,[1] Gerrad D. Jones[3]

**ABSTRACT** Despite the explosion of soil metagenomic data, we lack a synthesized understanding of patterns in the distribution and functions of soil microorganisms. These patterns are critical to predictions of soil microbiome responses to climate change and resulting feedbacks that regulate greenhouse gas release from soils. To address this gap, we assay 1,512 manually curated soil metagenomes using complementary annotation databases, read-based taxonomy, and machine learning to extract multidimensional genomic fingerprints of global soil microbiomes. Our objective is to uncover novel biogeographical patterns of soil microbiomes across environmental factors and ecological biomes with high molecular resolution. We reveal shifts in the potential for (i) microbial nutrient acquisition across pH gradients; (ii) stress-, transport-, and redox-based processes across changes in soil bulk density; and (iii) greenhouse gas emissions across biomes. We also use an unsupervised approach to reveal a collection of soils with distinct genomic signatures, characterized by coordinated changes in soil organic carbon, nitrogen, and cation exchange capacity and in bulk density and clay content that may ultimately reflect soil environments with high microbial activity. Genomic fingerprints for these soils highlight the importance of resource scavenging, plant-microbe interactions, fungi, and heterotrophic metabolisms. Across all analyses, we observed phylogenetic coherence in soil microbiomes—more closely related microorganisms tended to move congruently in response to soil factors. Collectively, the genomic fingerprints uncovered here present a basis for global patterns in the microbial mechanisms underlying soil biogeochemistry and help beget tractable microbial reaction networks for incorporation into process-based models of soil carbon and nutrient cycling.

**IMPORTANCE** We address a critical gap in our understanding of soil microorganisms and their functions, which have a profound impact on our environment. We analyzed 1,512 global soils with advanced analytics to create detailed genetic profiles (fingerprints) of soil microbiomes. Our work reveals novel patterns in how microorganisms are distributed across different soil environments. For instance, we discovered shifts in microbial potential to acquire nutrients in relation to soil acidity, as well as changes in stress responses and potential greenhouse gas emissions linked to soil structure. We also identified soils with putative high activity that had unique genomic characteristics surrounding resource acquisition, plant-microbe interactions, and fungal activity. Finally, we observed that closely related microorganisms tend to respond in similar ways to changes in their surroundings. Our work is a significant step toward comprehending the intricate world of soil microorganisms and its role in the global climate.

**KEYWORDS** soil microbiology, metagenomics, metaanalysis, machine learning, carbon cycling, nitrogen cycling, biogeochemistry, biogeography, soil microbiome, functional potential

S oil microbiomes catalyze some of the most biogeochemically important reactions on Earth, yet their complexity hampers our efforts to fully understand their function

Address correspondence to Emily B. Graham, emily.graham@pnnl.gov.

The authors declare no conflict of interest.

See the funding table on p. 15.

(1–6). Microbial destabilization of soil carbon stores, for instance, has positive feedbacks with global climate change (7–9), and microbial nutrient cycling sustains life at higher trophic levels (10–12). The molecular revolution, including advanced 'omics sequencing approaches, promises a new generation of fundamental and predictive understanding; imperatives for understanding how soil microbial communities individually and collectively contribute to carbon and nutrient cycling under environmental change. Yet despite the immense amount of soil metagenomic data now available, we lack a synthesized understanding of patterns in the distribution and functions of soil microorganisms that are critical to developing new and improved models (13–18).

While global patterns are well established for most macroorganisms, the extreme diversity of the soil microbiome and the myriad ecological forces that act upon it complicate efforts to understand patterns in its structure and function (19–27). Microbial assembly processes, dormancy, interspecies interactions, and aboveground-belowground connectivity in particular are among the many factors that influence soil microbial biogeography (19, 21, 26, 28–33). As a result, patterns in microbial distributions are almost always weaker than those observed in macroecological systems (26, 34, 35). Still, soil microbiome structure and function have been associated with variables such as latitude, vegetation, climate, and edaphic properties at local to global scales (3–5, 18, 19, 28, 34, 36–60).

A clear understanding of soil microbial biogeography is essential to predictions of soil microbiome function under novel climate scenarios. Microbiome composition (i.e., taxonomy and functional potential) impacts ecosystem processes (e.g., biogeochemical cycling) through differences in extracellular enzyme production, carbon use efficiency, symbioses, and other ecological traits among microorganisms (14, 44, 61–68). Correspondingly, changes in microbial distribution across environmental gradients can elucidate how soil functions may shift in relation to environmental change, particularly if novel ecosystems supplant existing soil environments (26, 69, 70). For instance, Ladau et al. (71) used current and historical biogeographical patterns of microorganisms to predict microbial distributions and possible ecosystem impacts under future climate scenarios.

New analytical techniques allow for the direct assessment of the potential functions encoded by soil microorganisms (4, 18), but genomic sequencing technologies also generate a tremendous amount of data. The detection of genes that are nearly ubiquitous (e.g., central metabolisms), poorly annotated, and/or of little relevance to soil processes in particular can obscure ecological patterns in the soil microbiome. Many researchers have attempted to distill this information into tractable units using trait-based approaches, keystone species analyses, and/or core microbiome assessments, with mixed results (27, 44, 62, 72–74). Other common challenges to understanding microbial biogeography include the detection of rare organisms and assigning thresholds for microbial taxonomy, both of which can influence patterns observed in soil microbiomes (34, 75).

As such, fundamental questions regarding global soil microbial distributions and their drivers remain largely unanswered (74, 76). Advanced analytics, including machine learning-based approaches, in combination with public sequence repositories are essential for the next generation of research in microbial ecology (18, 61, 77, 78). Here, we examine a comprehensive set of 1,512 manually curated soil metagenomic samples to decipher patterns in soil genomic potential across commonly measured environmental gradients and ecological biomes. We assay metagenomic sequences using an array of complementary annotation databases, read-based taxonomic assignment, and machine learning algorithms to extract multidimensional genomic fingerprints of global soil microbiomes. Our objective is to uncover never-reported biogeographical patterns in soil microbiome taxonomy and functional potential that can provide a molecular basis for predicting soil microbial responses to climate change.

## RESULTS

### Microbial clades and genomic potential across all soils

We collated soil metagenomic sequences from a wide range of environmental conditions across global biomes (Fig. 1; Tables S1 and S2). Bulk density, cation exchange capacity (CEC), nitrogen (N), pH, soil organic carbon (SOC), and clay content ranged from 0.28 to 1.56 $kg/dm^3$, 7.7 to 68.7 cmol C/kg, 0.25 to 22.4 g/kg, 4.4 to 8.3, 2.4 to 510.9 g/kg, and 2.7% to 57.1%, respectively.

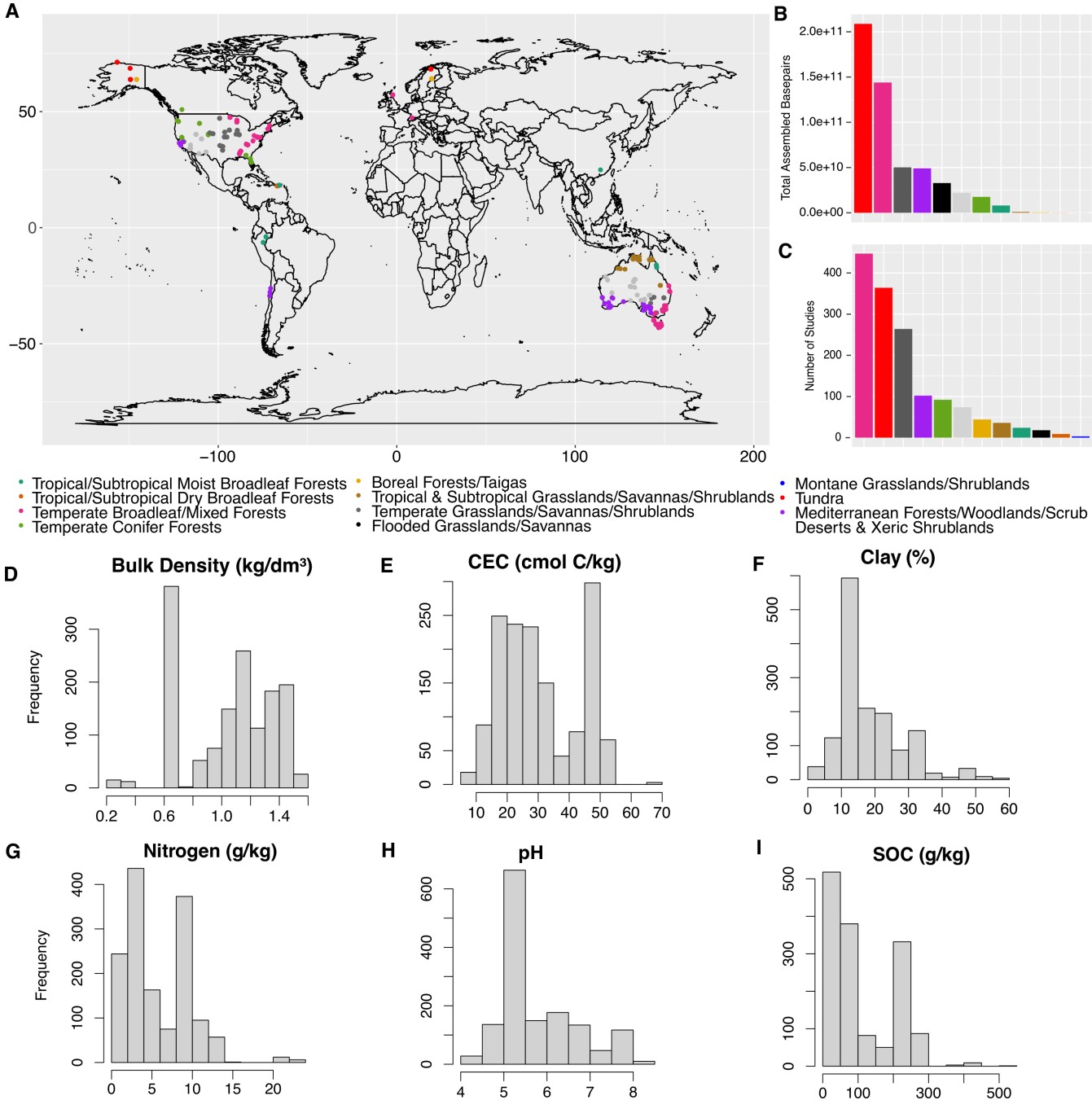

**FIG 1** Data set description. (A) Map of samples and distributions of (B) total assembled bp and (C) number of samples per biome according to Olson et al. (79). (D–I) Distribution of environmental variables across all soils from SoilGrids250m (80).

To reveal the full range of taxa and functional potential encoded by global soils, we first assessed microbial attributes [i.e., gene annotations by the Kyoto Encyclopedia of Genes and Genomes (KEGG) (81, 82), the Pfam (83) and TIGRFAM protein databases (84), and read-based taxonomy; see Materials and Methods] with the highest normalized abundance across all soils (Fig. 2). Multiple annotation databases provide complementary information and are a standard operating procedure by the U.S. Department of Energy Joint Genome Institute (JGI) (85, 86). Abundant KEGG orthologies (KO) were associated bacterial secretion systems (K12510), redox reactions (K00104, K00334, K03882, and K00380), and cellular growth processes (K02335, K02049, K00334, K03882, and K03088). Abundant Pfam were comprised of functions related to the tolerance of common soil stressors including temperature (PF11999, PF08496) and salinity (PF12838), as well as functions involved in microbial growth (PF08511, PF07357, PF13247, and PF00005). The most abundant TIGRFAM were also associated with soil stressors, for example, TIGR00229 (PAS Domain S-box protein) is involved in sensing changes in light, redox potential, oxygen, and other stressors (87). Additional highly abundant TIGRFAM across all soils included a sigma-factor that may be associated with microbial responses to environmental fluctuation (TIGR02937) and other signaling and sensing processes (TIGR00254, TIGR00231). Finally, abundant microbial genera included diverse microorganisms including Burkholderia, Mycobacterium, Streptomyces, Acetobacter, and Pseudonocardia.

## Variation in genomic potential across environmental gradients

To reveal biogeographic patterns in microbiomes across global soils, we extracted microbial attributes that were associated with soil factors and ecological biomes using random forest regression models (see Materials and Methods; Table S1). Soil pH, bulk density, and biome type were the strongest correlates of microbial attributes (i.e., they generated well-fitting models with comparatively high explanatory power relative to other factors; Table S1). The cross-validated random forest model for each of the variables above had a root mean square error of less than 0.8 (often <0.2) and/or at least 60% variance explained. Models for CEC, N, SOC, and clay content did not meet these

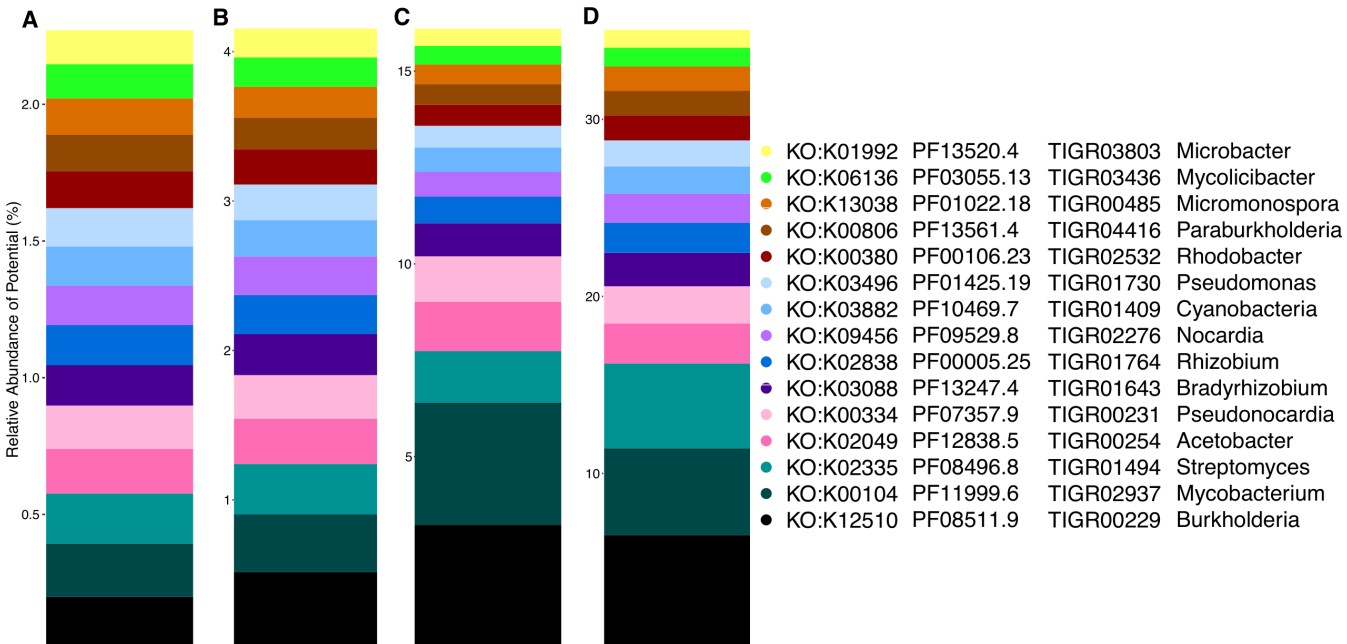

FIG 2 Highly abundant genomic attributes across all soils. Relative abundance of the top 15 most abundant soil microbial genomic attributes annotated by (A) KEGG orthology, protein family [(B) Pfam and (C) TIGRFAM], and (D) read-based taxonomy collated at the genus level.

thresholds. Results were conceptually consistent across different microbial attributes. A full description of all genomic fingerprints is in Extended Data S1.

pH separated soil microbiomes by nutrient- and vitamin B-related genes (Fig. 3). More alkaline soils were characterized by vitamin B-, nutrient-, and phosphatase-related activities, with genomic fingerprints that included K01662, PF01872, K08717, TIGR00378, PF01966, and PF00481 for instance, as well as many genera of Alphaproteobacteria. More acidic soils were signified by ammonia- and complex organic polymer- (e.g., wood-) related reactions, with genomic fingerprints that included K14333, K01684, TIGR03404, and TIGR02093, and microbial clades known to use ammonia as a substrate (Rhodopseudomonas and Klebsiella).

Bulk density separated soil microbiomes by genes associated with soil structure, hydrology, and nutrient content. The genomic fingerprint of low-bulk density soils comprised genes involved in organic N and methane cycling, as well as signatures of anoxia (Fig. 3). It included K00992, K11261, K11959, TIGR03323, TIGR01392, and microorganisms associated with anoxic and/or aquatic environments (Gardnerella, Cetia, Salmonella, Olleya, Mycoplasma, Escherichia, Enterobacter, and Chitinophaga). Conversely, high-bulk density soils were signified by genes involved in microbial transport, stress, pigmentation, and infection mechanisms. Notable attributes associated with high-bulk density soils included K00854, K11733, K03799, TIGR01263, TIGR01957, TIGR01203, PF00582, PF03631, and four Alphaproteobacteria.

Four biomes were highlighted with distinct genomic signatures by our models: (i) flooded grasslands and savannas, (ii) tundra, (iii) boreal forests, and (iv) montane grasslands/shrublands (Fig. 4). The soil genomic fingerprint of flooded grasslands/savannas was primarily characterized by positive associations with organic N (K13497, K07395, K19702, and K02042), phosphorus (P, K02042, K00854, K02800, and PF04273), stress tolerance (heat and arsenic: TIGR04282 and TIGR02348), and motility (PF00669, PF00482, and Paeniclostridium). The genomic fingerprint for tundra had the greatest number of positive gene associations, including those associated with methanogenesis (K11261, TIGR03323, TIGR03321, TIGR03322, TIGR03306, TIGR01392, and TIGR02145), nitric oxide production (PF08768), P transfer (K03525, K00997, K00992, K13497, and PF08009), and pathogenesis (PF11203, PF05045), supported by the presence of anaerobic (Gardnerella, Escherichia, Kosakonia, Glaesserella, Mycoplasma, Treponema, and Paeniclostridium) and possible plant growth-promoting genera (Kosakonia).

Boreal forests and montane grasslands/shrublands were primarily negatively associated with microbial attributes. Boreal forests displayed negative relationships with the normalized abundance of magnesium (Mg, K03284)- and P (K06217)-related genes and carbohydrate and organic N metabolism (K02800, K11472, PF00208, and PF04685), as well as some metal-related genes (TIGR00378, TIGR04282, and TIGR00003). Montane systems were negatively characterized by genes related to Mg and P (K01520, K04765, and TIGR01203), carbohydrate metabolism (K01816), virulence (PF09299), and motility (PF00669), as well as organic N (K07395, K00600, and PF00208) and S (K03644). Both boreal forests and montane grasslands/shrublands were negatively associated with anaerobic genera (Gardnerella, Thielavia, Coniochaeta, Escherichia, Salmonella, Kosakonia, Shigella, Mycoplasma, Treponema, Glaesserella, Enteractinococcus, Paeniclostridium).

Interestingly in all analyses across pH, bulk density, and biomes, we observed phylogenetic coherence in soil microbiomes—the normalized abundances of more closely related microorganisms tended to move congruently in response to the biome type and environmental factors (Fig. 4B; Extended Data S1).

## Unsupervised machine learning to disentangle environmental interactions associated with soil genomic potential

Finally, to detect emergent patterns in microbial biogeography, we used unsupervised machine learning to reveal a collection of soils with distinct genomic fingerprints. These soils and their resident microbiomes were associated with coordinated changes in SOC

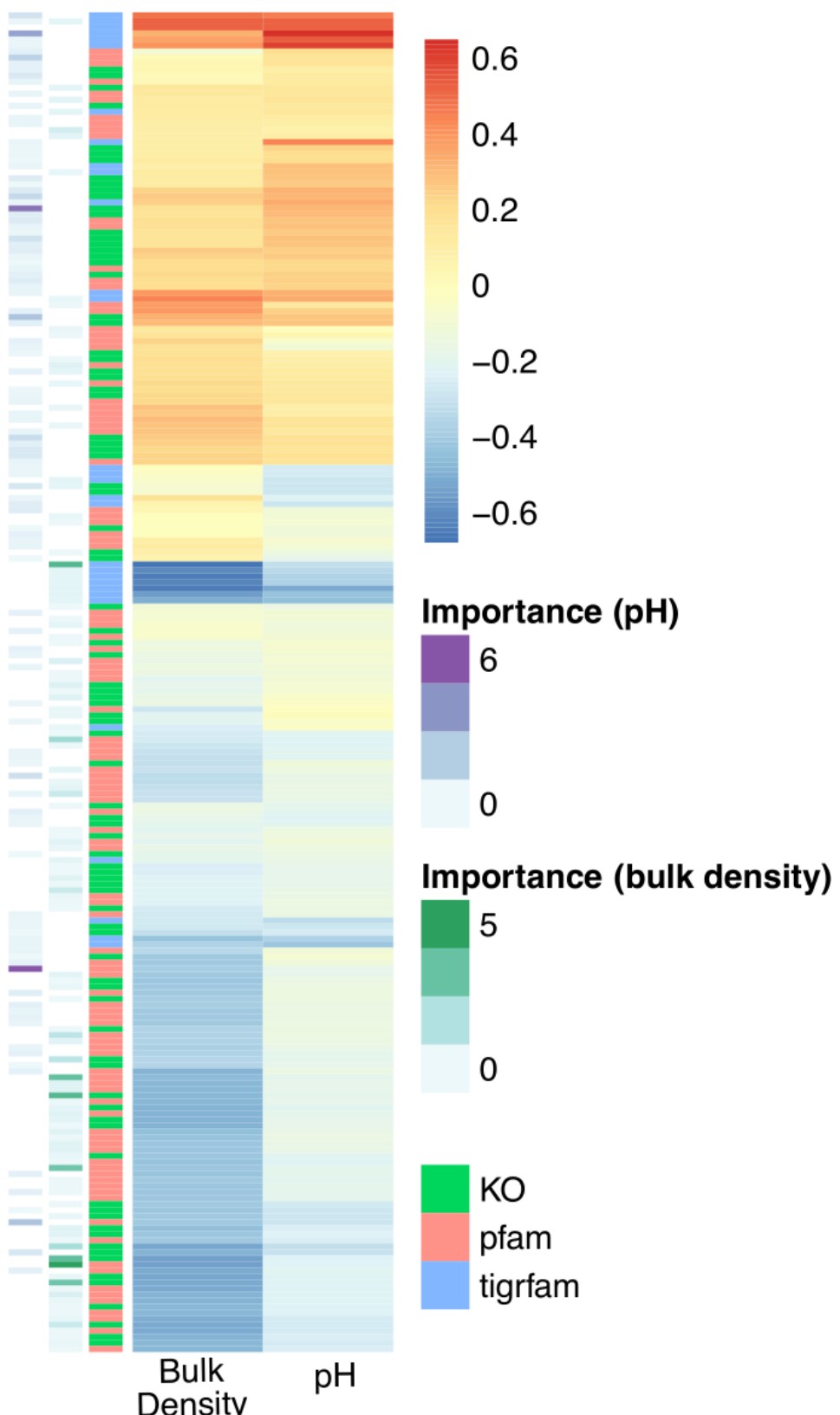

**FIG 3** Genomic fingerprints of high versus low pH and bulk density soils. Genomic attributes selected in at least one fingerprint for pH or bulk density are visualized. Pearson correlation coefficient is shown from blue to red in the primary heatmap, with bulk density correlations on the left and pH correlations on the right. The sidebars (respectively, from left to (Continued on next page)

**FIG 3** (Continued)

right) represent variable importance from random forest models of pH and bulk density and the type of attribute (e.g., KO, Pfam, and TIGRFAM). Please refer to Extended Data S1 for more information on the specific attributes associated with each soil environment. Read-based taxonomy collated at the genus level for each fingerprint is shown in Fig. 4.

content, total N, and CEC and with opposite coordinated changes in bulk density and clay content (Fig. 5; Extended Data S2). We identified KO cluster 7, genus cluster 6, and Pfam cluster 1 as containing soil samples that exhibited the strongest coordinated changes (defined by the number of significant correlations and $R^2$ values; see Materials and Methods). TIGRFAMs did not exhibit these patterns.

Selected microbial KO and genera were positively correlated with SOC, N, and CEC and negatively associated with bulk density and clay, while Pfam cluster 1 exhibited opposite patterns (i.e., negative associations with SOC, N, and CEC and positive associations with bulk density and clay). Soil microbiomes in KO cluster 7 and genus cluster 6 contained high normalized abundances of genes associated with energy generation (K00334, K00339, K00324, and K08738), filamentous bacteria/biofilms (K11903, Nocardia, Streptomyces, Mycobacterium, and Leptolyngbya), chemically complex organic matter decomposers (Nocardia, Rhodococcus, and Streptomyces), and N-fixing organisms and plant symbionts (Rhizobium, Bradyrhizobium, and Nostoc). Additionally, genes associated with N-cycling processes (K07685, K10041) and microbial interactions (e.g., signaling and secretion, K12537, and K02661) changed most dramatically across environmental gradients. The most variable taxa across environmental gradients were diverse—they included N-cycling (Uliginosibacterium, Microvirgula), autotrophic and/or anaerobic (Methanosalsum, Salana, Pseudobacteroides, Sulfurirhabdus, Microvirgula, Desulfallas, Pelotomaculum, and Risungbinella), and plant-related

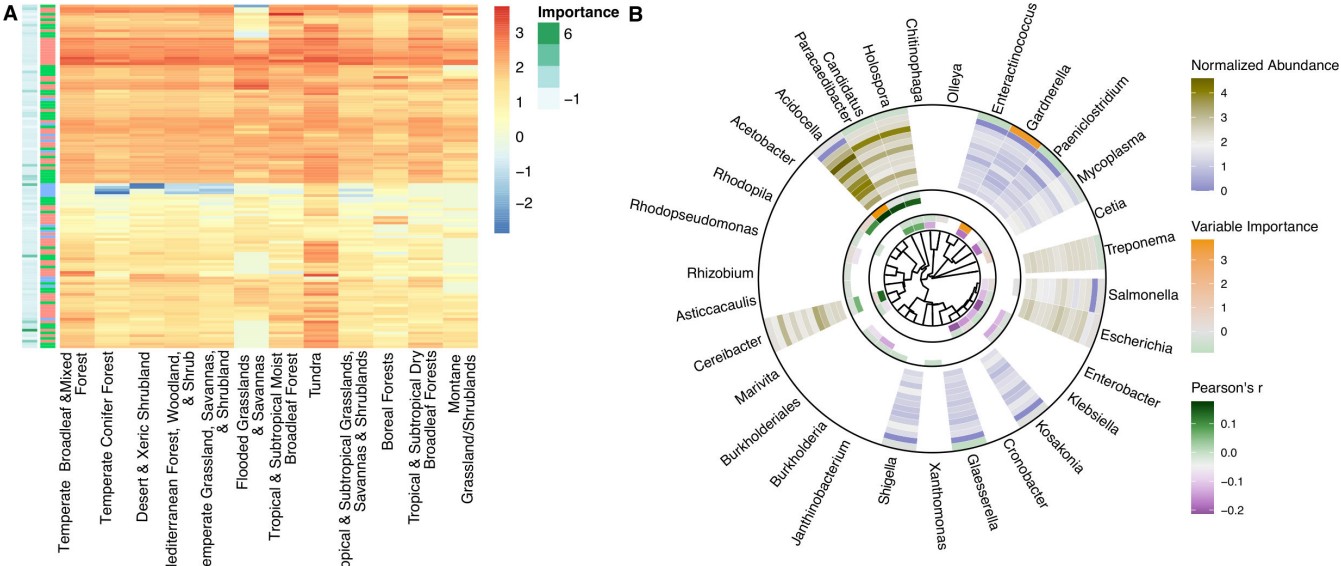

**FIG 4** Genomic fingerprints of soil biomes and phylogenetic distribution of selected soil taxa. (A) The normalized abundance of microbial attributes selected in at least one biome fingerprint is shown from blue to red. Variable importance is shown on the left side bar, and attribute type is shown on the right side bar. Please refer to Extended Data S1 for more information on the specific attributes associated with each soil environment. (B) Microbial genera selected in at least one fingerprint are depicted on a phylogenetic tree (generated by PhyloT). The innermost circle shows Pearson correlations with bulk density from green to purple and variable importance from green to orange. The middle circle shows Pearson correlations with pH from green to purple and variable importance from green to orange. The outer circle shows the normalized abundance of genera across biomes (from outer to inner: temperate broadleaf and mixed forests; tundra; Mediterranean forests, woodlands, and scrub; tropical and subtropical grasslands, savannas, and shrublands; temperate grasslands, savannas, and shrublands; boreal forests/taigas; deserts and xeric shrublands; temperate conifer forests; tropical and subtropical dry broadleaf forests; flooded grasslands and savannas; tropical and subtropical moist broadleaf forests; and montane grasslands and shrublands). Variable importance is shown in the outermost ring from green to orange.

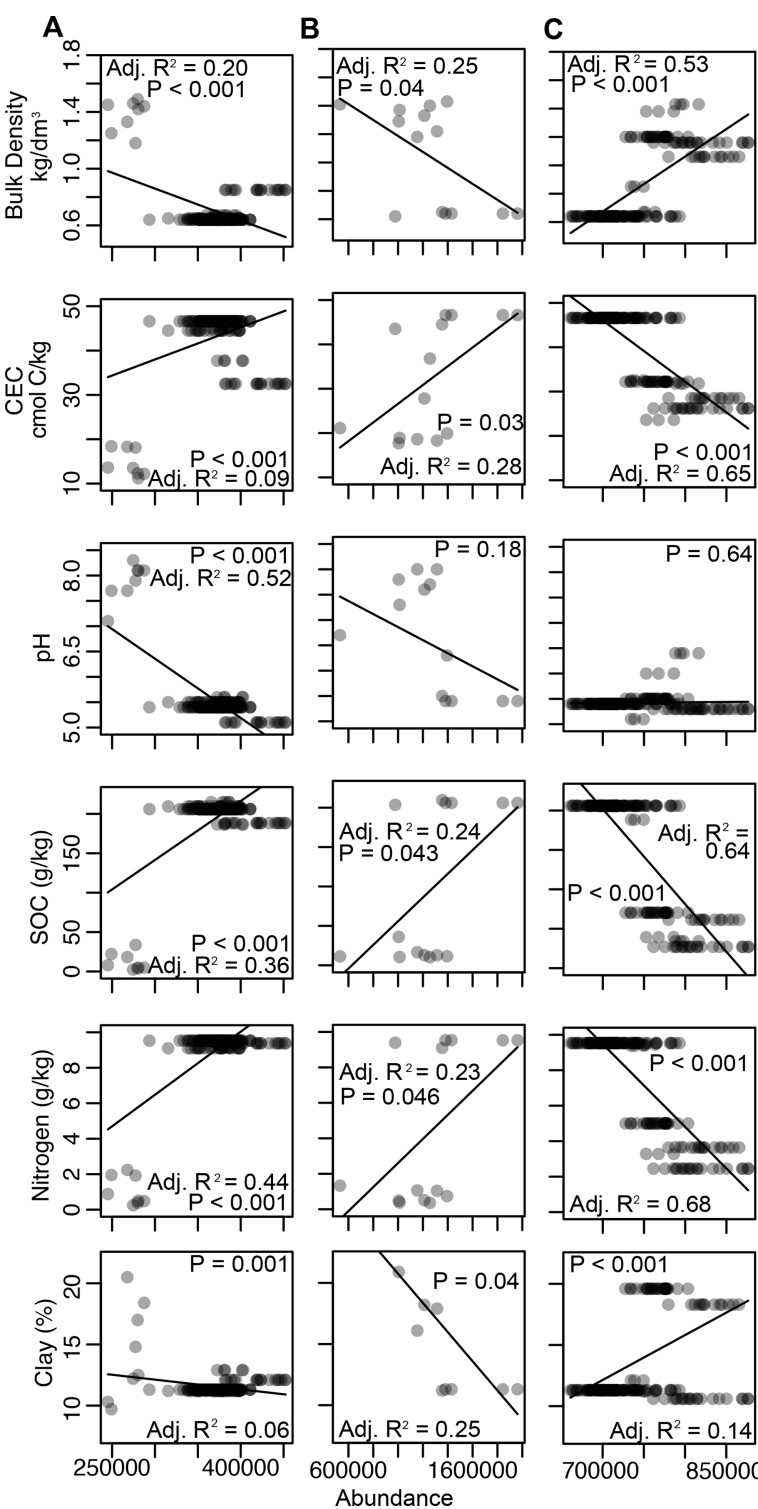

**FIG 5** Unsupervised clusters of microbial attributes associated with multiple soil factors. (A) KEGG orthology, (B) microbial genus, and (C) Pfam. TIGRFAM annotations did not yield a satisfactory model. Rows represent environmental variables. The total normalized abundance of an attribute per sample is plotted against each environmental variable. Lines and statistics represent linear regression.

microorganisms (Pseudobacteroides, Zeaxanthinibacter, and Parapedobacter). Samples in Pfam cluster 1 had high relative abundances of organic N- (PF01979, PF08352, and PF01842), P- (PF00406, PF01380), and signaling-related (PF01627, PF09413) genes, while

many genes thought to be associated with eukaryotes changed most dramatically across environmental gradients in Pfam annotations (PF04178, PF03188, and PF01793).

## DISCUSSION

### Microbial taxa and functional potential in global soils

Across all soils, the most abundant microbial attributes spanned disparate lineages and potential functions, highlighting the high microbial diversity of soils relative to other global habitats (5, 18, 45, 46). While we note that we assayed microbial genes which do not necessarily reflect expressed functions (61, 88–91), the genomic potential revealed by this investigation provides a basis for more targeted research into soil microbiome function across different soil habitats. The most prevalent microbial genera included members of commonly observed soil taxa including Alphaproteobacteria, Betaproteobacteria, and Actinobacteria (45). Widespread genomic functions included growth, resource acquisition, and stress tolerance strategies that are essential for microbial persistence in heterogeneous soil environments. Bacterial secretion systems were common, as well as DNA repair/replication, transport, and energy generation functions. Interestingly, genes for the sensing and/or tolerance of fluctuations in temperature, salinity, light, and redox conditions were among the most abundant soil microbial genes; all of which are common soil microbial stressors (92, 93). Collectively, the diverse taxa and functions encoded by the soil microbiome underscores the heterogeneous stressors on the soil microbiome. The high global soil microbial diversity observed here can obscure patterns in microbiome structure and function, indicating the use of machine learning-based tools to unravel its complexity (18, 45, 88).

### Soil genomic potential corresponds to pH, bulk density, and biome type

Soil genomic patterns were evident across soil pH, bulk density, and biomes; however, some hypothesized drivers of the soil microbiome including SOC, total N, and CEC were not statistically linked to microbial taxa or genomic functional potential. Soil carbon, N, and cation exchange capacity can reflect resource availability and are often associated with soil microbiome change at local scales (14, 36, 89–91). The lack of trends at the global scales may represent heterogeneity in the importance of resources in structuring microbiomes; indeed, the influence of deterministic (e.g., selection) versus stochastic (e.g., random) assembly processes on soil microbiomes varies through space and time (30, 94–97). While our results support pH and biome type as primary determinants of soil microbial structure and function (5, 6, 28, 45), it is interesting that bulk density was also a strong correlate of genomic patterns. This may reflect the importance of biophysical and hydraulic properties on soil microbiome structure and function (94, 98–100).

Collectively, we reveal novel biogeographical patterns in specific aspects of soil microbiome composition and potential function that have not been reported elsewhere. Our results are consistent with ecological processes hypothesized to influence the soil microbiome and provide greater molecular specificity than that revealed by previous works. They enable us to better understand microbial lifestyles that support the persistence of certain members over others; such knowledge underlies a predictive understanding of soil microbiome changes with shifts in the global climate.

### Microbial nutrient acquisition potential shifts across pH gradients

The genomic fingerprint of alkaline soils demonstrated an association with microbial nutrient acquisition and is important for understanding belowground microbial dynamics in arid and semiarid ecosystems [>35% of global land surface (101)]. The soil matrix in these systems is porous, has low rates of water retention, and contains an abundance of calcium carbonate that limits P bioavailability (102). Accordingly, the alkaline soil fingerprint was enriched in genes encoding phosphatases and related functions (PF01966, PF00481) that may signify the importance of microbial P acquisition and/or limitation in alkaline soils (103–106). Vitamin B-related genes were also included

in this fingerprint and have been linked to P solubilization (K01662, PF01872) (107). The presence of a urea transporter (K08717) may likewise indicate the importance of nutrient acquisition in alkaline soils (108). Notably, the high-$Ca^{2+}$ alkaline soil environment was also characterized by a calcium/proton exchanger (TIGR00378)—which regulates intracellular $Ca^{2+}$ concentrations to prevent $Ca^{2+}$ overload (109) and an uncharacterized hydrophobic domain (TIGR00271) that may arise from low-moisture environments that are often characteristic of alkaline soils.

In contrast, more acidic soils were characterized by a distinct set of nutrient acquisition genes, in particular genes encoding ammonia and wood decomposition reactions that produce $H^+$ as a byproduct. For instance, nitrification is a hallmark of microbial genera Rhodopseudomonas and Klebsiella (110–112) and increases soil acidity through the release of $H^+$ from ammonia. Likewise, wood-degrading fungi often secrete acidic compounds to aid in the digestion of chemically complex polymers (113–116). This is consistent with the association of K14333, K01684, TIGR03404, and TIGR02093 with low-pH soils.

Finally, Alphaproteobacteria have been previously shown to be sensitive to variation in soil pH (117) and were included in the genomic fingerprints of both high- and low-pH soils, further bolstering their potential sensitivity to global change processes that impact soil pH (e.g., atmospheric nitrogen deposition).

## Genes encoding stress, transport, and/or redox-based processes change with bulk density

Bulk density reflects soil biophysical properties such as texture, water content, porosity, and mineralogy that drive microbial community structure and function (19, 53, 54, 118, 119). Patterns in the relationships between these variables and the soil microbiome can be complex at global scales (19), and our findings are to our knowledge the first to report a global-scale relationship between bulk density and the distribution of genes encoded by the soil microbiome.

Low-bulk density soils were characterized by genes common to water-logged and/or agricultural soils, while high-bulk density soils included potential functions associated with transport mechanisms, stress tolerance, and virulence. For instance, K11261, TIGR03323, and TIGR01392—all signatures of low-bulk density soil—are each associated with methanogenic processes that are prevalent in saturated soils (120). Low-bulk density soils were further characterized by anaerobic and/or aquatic microorganisms including Gardnerella, Cetia, Salmonella, Olleya, Mycoplasma, Escherichia, Enterobacter, and Chitinophaga that also denote lifestyles associated with microbial persistence in water-saturated environments (121–129). Though we did not directly assess land cover, the presence of potential functions associated with organic N cycling in the low-bulk density fingerprint (K00992, K11959) may additionally suggest the influence of agriculture on soil microorganisms, where tillage generates loosely packed soils (i.e., low bulk density) and fertilizer application enhances nutrient cycles.

In contrast, the genomic fingerprint of high-bulk density soils suggests an environment inclusive of density-dependent relationships between microorganisms, their substrates, and pathogens (transport: K00854, K11733, TIGR01957, and TIGR01203; virulence: PF03631 and TIGR01203). Stress-related genes (K03799, PF00582.24) may be reflective of this competitive environment or related to ecosystem processes such as soil compaction which is common in deserts (K03799).

## Soil genomic potential for greenhouse gas emissions differs by biome

Finally, the soil habitat (reflected here by biome type) seemed to be a driver of dissimilarity in microbiome structure and potential function, providing detailed information on the molecular biology of global soil ecosystems.

We first highlight microbial genomic attributes of tundra ecosystems, as they are experiencing rapid transformations in response to global climate change (130, 131). Tundra exhibited the highest number of microbial taxa and potential functions in their

genomic fingerprint, the vast majority of which displayed positive associations. Given the relationship between soil microbial diversity and biogeochemical function, this provides a consistent and distributed genomic basis for the enhanced greenhouse gas emissions that have been associated with permafrost thaw in tundra (130, 132, 133). The microbial attributes associated with tundra in this study reveal clues into the biology that drives active biogeochemical cycles in these vulnerable ecosystems. Methanogenesis in particular is responsible for fluxes of methane in warming tundra (132, 134, 135) and was associated with tundra in the current study. Additionally, a gene involved in nitric oxide cycling (PF08768) (136–138), as well as a variety of anaerobic organisms and P-related genes (K03525, K00997, K00992, K13497, and PF08009) (139–141), also provide insight into the molecular functions of tundra in the context of global climate change. Interestingly, we also found evidence for pathogen resistance within the tundra genomic fingerprint (PF11203, PF05045), lending credence to a small number of recent studies suggesting permafrost environments as one of the largest reservoirs of soil viruses (142–145) and a possible linkage between soil methane and viral infection (146).

Flooded grasslands/savannas were also generally positively associated with many microbial attributes, several of which encoded nutrient cycling, stress tolerance, and/or motility traits. Genes related to organic N cycling (K13497, K07395, K19702, and K02042), P (K02042, K00854, K02800, and PF04273), and/or stress tolerance (TIGR04282, TIGR02348) may signify microbial adaptations to nutrient scarcity and other stressors. Motility (PF00669, PF00482, and Paeniclostridium) may be reflective of beneficial microbial lifestyles in saturated systems, where movement is facilitated by high pore space connectivity. Interestingly, a mechanism for arsenic resistance was affiliated with flooded grasslands and savannas, possibly denoting arsenic groundwater contamination in many regions of the world (147).

In contrast, boreal forests and montane grasslands/shrublands were signified by negative associations with Mg- and P-related genes (boreal: K03284 and K06217; montane: K01520, K04765, and TIGR01203) and with organic N- and carbohydrate-related genes (boreal: K02800, K11472, PF00208, and PF04685; montane: K01816, K07395, K00600, and PF00208). In addition, anaerobic microbial genera were negatively associated with these ecosystems. The boreal forest fingerprint uniquely included some metal-related genes (TIGR00378, TIGR04282, and TIGR00003), while montane systems were uniquely characterized by virulence (PF09299), motility (PF00669), and sulfur-related genes (K03644). The negative associations with carbon, N, and P suggest that carbon and inorganic nutrients may be more available in these systems relative to other biomes; and Mg is often associated with plant growth (148). Taken together, microbial fingerprints of boreal forests and montane grasslands/shrublands convey slower growth rates and fewer resource constraints than other ecosystems.

## Resource scavenging, plant-microbe interactions, fungi, and heterotrophic metabolisms prevalent in fertile soils

Relationships between biomes and specific microbial attributes suggest that there may be multiple simultaneous factors that structure soil microbiome composition and function. Because of this, we next explored relationships between microbial attributes and soil factors that arose naturally out of unsupervised analysis. This allowed us to infer interactive effects that could not be assessed when specifying a single environmental gradient.

Unsupervised machine learning uncovered soil microbial genomic attributes associated with coordinated changes in SOC, total N, and CEC and oppositely with clay content and bulk density that may represent some of the most biologically active soils on Earth (Fig. 5). Soils rich in SOC and N with high CEC constitute resource- and energy-rich environments, supported by an abundance of energy-generating genes in the fingerprint of these soils (K00334, K00339, K00324, and K08738). Interestingly, these environments are best explained by a combination of environmental variables rather than an individual factor. As an example, soils with high SOC alone could be limited by

N or other nutrients; thus, we propose that genomic fingerprints for multiple simulta-neously interacting factors may be among the most critical for understanding global biogeochemistry. They are nearly impossible to discern without unsupervised statistical approaches, and we know of no other work that has done so.

Soils with high SOC, N, and CEC were characterized by genes associated with resource scavenging, plant-microbe interactions, and heterotrophic metabolisms. We observed numerous attributes in their genomic fingerprint associated with filaments or biofilms, intra-organism signaling, and/or chemically complex organic matter decompo-sition (K11903, K12537, K02661, Nocardia, Streptomyces, Mycobacterium, Leptolyngbya, Nocardia, Rhodococcus, and Streptomyces). This is consistent with microbial morpholo-gies and lifestyles that support the decay of chemically recalcitrant polymers (e.g., wood), as well as nutrient acquisition and transport across soil pore networks. Plant-microbe interactions, including symbiotic N-fixing organisms, were also a key feature of these soils, highlighting the importance of above- and belowground connectivity in pro-ductivity (Rhizobium, Bradyrhizobium, Nostoc, Pseudobacteroides, Zeaxanthinibacter, Parapedobacter, and Zea) (149). In contrast, anaerobic and autotrophic organisms were highly sensitive to changes across these connected environmental gradients, possibly reflecting their exclusion from more oxygen- and energy-rich environments where heterotrophic organisms tend to dominate.

We also selected one Pfam cluster that was associated with low SOC, N, and CEC and with high clay and bulk density soils, which seemed to underscore the importance of fungi in resource- and energy-rich environments. The most sensitive genes in this cluster of samples corresponded to eukaryotic organisms (PF04178, PF03188, and PF01793). Previous work has shown fungal sensitivity to soil resource availability, as well as high fungal diversity in organic-rich tropical soils (52). Overall, these insights provide greater resolution into the microbial mechanisms that support both belowground biogeochemi-cal cycles and aboveground productivity than ever before, and they can provide a basis for the next generation of microbial ecology research and model development.

## Phylogenetic coherence in microbial biogeography

The extent to which selection acts on phylogenetically conserved properties of soil microbiomes and further translates into differences in functional potential [e.g., response-effect traits (150)] is a long running unknown in microbial ecology. Some previous studies have shown little connection between functional attributes and microbial phylogeny (44, 151), while others have indicated the existence of taxa-function relationships (88). And still others have suggested variation in the coupling of taxonomy and function across various functions of interest, levels of phylogenetic resolution, and/or spatial scales (34, 55, 61, 152, 153). Though microbial taxa are generally less correlated to the environment than functional potential (Table S1), we found strong phylogenetic coherence in the response of genera to changes in the soil environment (Fig. 4B). That is, closely related organisms tended to respond similarly to environmental change. This implies that evolutionary history, life history strategies, and/or morphology [all traits that tend to be phylogenetically conserved (154)] could influence changes in microbial community membership in response to environmental change. As such, metabarcoding approaches that yield microbiome taxonomy may be among the first indicators of ecosystem transitions, such as response to disturbances and transitions between state archetypes (30).

## Conclusions

Despite the significance of soil microorganisms in global biogeochemistry, we still know little about the ecological processes that regulate their community composition and function across the wide range of global soil environments. We used machine learning to sift through tens of thousands of taxa and potential functions encoded by 1,512 global soil microbiomes, revealing novel biogeographical patterns in soil microbiome composition and functional potential with high molecular resolution. Specifically, we

demonstrate shifts in the potential for (i) microbial nutrient acquisition across pH gradients; (ii) stress, transport, and redox-based processes across changes in soil bulk density; and (iii) genes and organisms associated with soil greenhouse gas emissions across biomes. We also uncover a suite of metabolic processes that are enriched in the microbial genomes of energy-rich soils. These changes were coincident with phylogenetically congruent compositional shifts—suggesting that closely related soil microbial taxa are sensitive to similar environmental stressors. Our work enables us to better understand microbial lifestyles that support the persistence of certain members over others along environmental gradients; such knowledge underlies a predictive understanding of soil microbiome changes with shifts in the global climate and is vital to constraining biochemical reactions that regulate the release of greenhouse gasses from soils.

## MATERIALS AND METHODS

### Data set description and normalization

We collected 1,512 manually curated soil metagenomic samples available in the Joint Genome Institute's (JGI) Integrated Microbial Genomes and Microbiomes (IMG/M) platform in August 2020 as described by Graham et al. 2024 (accepted), of which 1,493 were associated with latitudinal and longitudinal information. Samples span most major biomes and continents (Fig. 1; Tables S2 and S3). Sequences were quality controlled, assembled, and annotated by KEGG Orthology (KO), the Pfam and TIGRFAM protein family databases, and read-based taxonomy by the JGI's standard workflows (82–86). While sample collection and sequencing methods inevitably vary in metaanalyses such as the current study, we applied standardized analytical workflows to minimize methodological artifacts to the greatest extent possible. Due to the collective size of assembled sequences, we focused on annotations and read-based taxonomy for downstream analysis, which is more computationally feasible than sequence assemblies or metagenome-assembled genomes that are impractical at the scale of thousands of samples.

Each data set (i.e., KO, Pfam, TIGRFAM, and read-based taxonomy) was independently normalized by the following procedure prior to analysis. For Pfam and KO annotations only, samples with greater than 2,000 total reads were retained. For TIGRFAM annotations only, samples with greater than 500 total reads were retained. A lower cutoff was used for TIGRFAM data as the total read counts were lower in this data set. For all three annotations, only functions with a non-zero value in at least half of the samples or a total count of 7,560 (corresponding to an average of 5 reads per sample for each function) were retained. For microbial genera, we retained only samples with greater than 4,000 counts and genera with a non-zero value in at least half of the samples or a total count of 7,560 (corresponding to an average of 5 reads per sample for each genus). After removing low-abundance samples and functions/genera, all four datasets were then normalized using the trimmed mean of M values method (155). We found that this method gave us the best results when examining normalized data using a box and whisker plot (Figure S1).

We assigned the biome type and soil properties to each sample using publicly available resources. We used latitude and longitude to derive using Olson biomes (79) for each sample using data provided by the World Wildlife Federation (156). Soil parameters were collected from the SoilGrids250m database from 0 to 5 cm (80). SoilGrids250m is a spatial interpolation of global soil parameters using ~150,000 soil samples and 158 remote sensing-based products. Here, we focus on six parameters often associated with soil microbiomes: bulk density, cation exchange capacity, nitrogen, pH, soil organic C, and clay content. Because our focus on spatial dynamics and soils were collected at various times, we did not include temporally dynamic variables such as soil moisture or temperature in our set of environmental parameters, though we acknowledge they may have profound impacts on the soil microbiome. Additionally, because soil data are derived from a comprehensive spatial interpolation and not measured directly on each

soil sample, our models may fail to capture local-scale heterogeneity in relationships. Further investigations that generate and leverage fully standardized data are needed to resolve this discrepancy. Nonetheless, local heterogeneity should statistically weaken the relationships observed here, and our results present a promising investigation into the global biogeography of soil microbiomes.

## Machine learning for genomic fingerprints

We selected random forests as our primary statistical approach by assessing the ability of various machine learning algorithms to adequately parse soil microbial attributes across environmental variables and ecological biomes. Models were built and validated using 10-fold cross-validation (CV) repeated five times, 50% of the data for training, and 50% of the data for testing. Models from different algorithms were compared using CV scores and variance explained (Figure S2). Random forest models consistently yielded comparatively high CV scores and proportion of variance explained.

Thus, we constructed random forest models to extract genomic fingerprints from soil microbiomes using two approaches: (i) a supervised split across variables known to be associated with soil microbiomes and (ii) an unsupervised approach that allowed predictors and their interactions to naturally arise. Random forest models were built in R software (157) for each attribute type independently using the "caret" package (158).

In approach i, we divided samples into either two groups containing the highest and lowest 10% of values for each environmental factor or one group per biome. We then used random forests to rank each microbial attribute according to their importance (i.e., difference in mean square error normalized by standard error, "varImp" function in "caret" package) to differentiate samples across each environmental factor or biome. Subsequently, because variable importances followed an exponential decay, we used breakpoint analysis to define a set of the most important variables distinguishing each group ("strucchange" package, $h = 0.01$ with a minimum segment size of two) (159). Because this resulted in one set of attributes that drove variation in soil microbiomes in relation to a given environmental factor or biome, we extracted attributes associated with high (positive slope) or low (negative slope) values of each environmental factor using linear regression. The resultant attributes comprised two genomic fingerprints per environmental factor or biome—one associated with low values and one associated with high values.

In approach ii, we first used k-means clustering to group samples with similar microbial attributes using R packages: "vegan" (160), "mgcv" (161), and "cluster" (162). The optimal number of clusters was chosen by evaluating the total sum of squares and the silhouette index (163). To determine the extent to which soil microbiomes within each cluster corresponded to changes in environmental factors, we related the total normalized abundance of attributes per sample to each environmental factor using linear regression. We then selected one cluster per attribute type for deeper investigation, chosen by the number of significant correlations with environmental factors. To generate a genomic fingerprint for the selected cluster, we extracted attributes that were either most abundant or most variable within the cluster. As in approach i, we used breakpoint analysis to define a set of the most abundant attributes per cluster ("strucchange" package, $h = 0.01$ with a minimum segment size of two). To determine the most variable attributes per cluster, we calculated the coefficient of variation for each attribute and then used breakpoint analysis to extract the most variable attributes.

We used the following packages for data manipulation and visualization: "ggplot2" (164), "dplyr" (165), "factoextra" (166), "Hmisc" (167), "colorspace" (168), "RColorBrewer" (169), "gridExtra" (170), "tidyverse" (171), and "maps" (172). Additionally, for visualizations of read-based taxonomy, we generated phylogenetic trees using phyloT v2 (173), an online tree generator based on the Genome Taxonomy Database. Then, we visualized the tree in R using the packages "ggtree" (174), "treeio" (175), and "ggnewscale" (176). We generated heatmaps using the "pheatmap" package (177).

## ACKNOWLEDGMENTS

This work was performed under a Laboratory Directed Research and Development (LDRD) from the Pacific Northwest National Laboratory (PNNL). PNNL is operated by the Battelle Memorial Institute for the U.S. Department of Energy under Contract No. DE-AC05-76RL01830. We thank the U.S. Department of Energy Joint Genome Institute (JGI) for maintaining the public Integrated Microbial Genomes and Microbiomes (IMG/M) platform from which we obtained this data set.

## AUTHOR AFFILIATIONS

[1]Earth and Biological Sciences Directorate, Pacific Northwest National Laboratory, Richland, Washington, USA
[2]School of Biological Sciences, Washington State University, Pullman, Washington, USA
[3]Department of Biological and Ecological Engineering, Oregon State University, Corvallis, Oregon, USA

## AUTHOR ORCIDs

Emily B. Graham  http://orcid.org/0000-0002-4623-7076
Vanessa A. Garayburu-Caruso  http://orcid.org/0000-0003-3383-6237
Ruonan Wu  http://orcid.org/0000-0001-9466-4462
Jianqiu Zheng  http://orcid.org/0000-0002-1609-9004
Ryan McClure  http://orcid.org/0000-0003-0573-6917
Gerrad D. Jones  http://orcid.org/0000-0002-1529-9506

## FUNDING

| Funder | Grant(s) | Author(s) |
|---|---|---|
| DOE | SC | Pacific Northwest National Laboratory (PNNL) | | Emily B. Graham |

## AUTHOR CONTRIBUTIONS

Emily B. Graham, Conceptualization, Data curation, Formal analysis, Funding acquisition, Investigation, Methodology, Project administration, Resources, Software, Supervision, Validation, Visualization, Writing – original draft, Writing – review and editing | Vanessa A. Garayburu-Caruso, Data curation, Formal analysis, Investigation, Methodology, Writing – review and editing | Ruonan Wu, Data curation, Formal analysis, Investigation, Methodology, Supervision, Writing – review and editing | Jianqiu Zheng, Conceptualization, Funding acquisition, Methodology, Supervision, Writing – review and editing | Ryan McClure, Conceptualization, Data curation, Formal analysis, Funding acquisition, Investigation, Methodology, Supervision, Writing – review and editing | Gerrad D. Jones, Conceptualization, Data curation, Formal analysis, Funding acquisition, Investigation, Methodology, Supervision, Writing – review and editing

## DATA AVAILABILITY

All data are publicly available via the JGI IMG/MG platform. Please see Tables S2 and S3 for further information. Code for data analysis and all figures is available at 10.6084/m9.figshare.25374427.

## ADDITIONAL FILES

The following material is available online.

## Supplemental Material

**Extended Data S1 (mSystems01112-23-s0001.xlsx).** Supervised fingerprints.
**Extended Data S2 (mSystems01112-23-s0002.xlsx).** Unsupervised fingerprints.
**Figure S1 (mSystems01112-23-s0003.pdf).** Abundances across different normalization procedures.
**Figure S2 (mSystems01112-23-s0004.pdf).** Comparison of machine learning algorithms.
**Legends (mSystems01112-23-s0005.docx).** Legends for Figures S1 and S2.
**Table S1 (mSystems01112-23-s0006.xlsx).** Model performance for random forest regressors.
**Table S2 (mSystems01112-23-s0007.csv).** Data sets and associated sequence metadata.
**Table S3 (mSystems01112-23-s0008.csv).** Geographic metadata for each sample.

## Open Peer Review

**PEER REVIEW HISTORY (review-history.pdf).** An accounting of the reviewer comments and feedback.

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
