## [Reviewer comments · mSystems]

Genomic fingerprints of the world's soil ecosystems

Emily Graham, Vanessa Garayburu-Caruso, Ruonan Wu, Jianqiu Zheng, Ryan McClure, and Gerrad Jones

Corresponding Author(s): Emily Graham, Pacific Northwest National Laboratory

Review Timeline:

Submission Date:	October 18, 2023
Editorial Decision:	November 13, 2023
Revision Received:	March 11, 2024
Accepted:	March 25, 2024

Editor: Ashley Shade

Reviewer(s): Disclosure of reviewer identity is with reference to reviewer comments included in decision letter(s). The following individuals involved in review of your submission have agreed to reveal their identity: Zachary B Freedman (Reviewer #2)

Transaction Report:

DOI: <https://doi.org/10.1128/msystems.01112-23>

Re: mSystems01112-23 (Genomic fingerprints of the world's soil ecosystems)

Dear Dr. Emily B Graham:

Editor comment: Thanks for your submission! Both reviewers were positive, and so we provide a request for a minor modification on your work. Please address their comments in full, and we reserve the option to request reviewer feedback on the author response.

In the revision, please include a link to the annotated and reproducible computational code for the meta-analysis and a statement of this availability within the Data Availability section. This can be in GitHub, FigShare, as a supplemental file, or wherever your team typically shares open code.

Revision Guidelines

Sincerely,
Ashley Shade
Editor
mSystems

Reviewer #1 (Comments for the Author):

This study aims to reveal the distribution of the global soil microbiome based on 1512 soil metagenomic data. The topic is interesting as it helps us to understand the distribution and function of soil microbiomes and how they could change under scenarios of global change. However, the draft needs further improvement, and I have specific comments that need to be addressed.

1. Line 17, should specify the main objectives of the study, given that there are already patches of work addressing this topic.
2. Line 19, why mention the release of greenhouse gases here?
3. Lines 23-25 are hard to understand and require clarification.
4. Line 50 should state that almost every aspect of the soil biogeochemical process is affected, not just "some."
5. Line 67 should specify which factors matter.
6. I strongly recommend that the authors focus more on the logical flow of the introduction, highlight the current knowledge gap, and explain how this study improves the knowledge of the readers rather than doing a literature synthesis.
7. Lines 105-106 require clarification.
8. After reading the results section, the main findings and the story that the authors want to express are not clear.
9. Line 206, the discussion section is too long, and I suggest condensing it and focusing on the main story and ecological implications.
10. Line 406 should address how to minimize any possible bias in meta-analysis, particularly regarding soil sampling and sequencing processes.
11. Lines 429 and 447, results should be included, at least in the supplementary section.
12. Line 443, data were generated from global prediction and not observation data, raising concerns about the accuracy of the results.
13. Line 445 would benefit from a confusing matrix to understand the prediction power of this random forest model.
14. While the current results are mainly about the composition of soil microbiomes, metagenomic data mining should also include how functional genes and protein families distribute and change with environmental gradients.

Reviewer #2 (Comments for the Author):

In this manuscript, the authors synthesize > 1500 metagenomes across continents and diverse ecosystems to determine the macroecology of soil microbial genomic potential and its ecological controls. Using both supervised and unsupervised random forest, the authors determine distinct genomic fingerprints of soil types and connect plausible mechanisms by which particular functional gene groups may relate to particular soil characteristics. Overall, this paper is sound and represents an impressive effort with a very large dataset to address a timely knowledge gap in microbial macroecology. I don't have any major critiques, but I think the manuscript would benefit from a dedicated discussion paragraph about the limitations of this effort/analysis.

For example, one limitation to discuss relates to In lines 435-436, where it seems like soil data like bulk density, cation exchange CEC, nitrogen, etc were not from the samples as sequenced by JGI but from GIS grids/pixels. I appreciate that temporally dynamic variables were not included, but some of these variables included can vary across a landscape, and this degree of variance may also differ by ecosystem type. For example, could the strength of associations between bulk density, pH, and genomic potentials and the lower strength of associations between genomic potentials with N and CEC be due to potentially greater landscape scale variance in N and CEC versus less landscape variability bulk density and pH? Similarly, given conclusions about uncovering "microbial mechanisms underlying soil biogeochemistry", it would be helpful to discuss the limitations regarding the potential lack of coherence between functional gene abundance and biogeochemical function should be included.

Minor comments:

Line 255: Consider either expanding this paragraph or deleting it. It could be expanded by highlighting some examples of specific taxa at the genus or family level that are associated with vitamin B processes.

Line 274: are most of the high bulk density samples agricultural? If so, would be good to mention here.

Line 291: This sentence doesn't seem supported by data. Would be helpful to remind reader here about the result indicating associations between genes associated with methanogenesis and tundra samples.

Reviewer #1 (Comments for the Author):

This study aims to reveal the distribution of the global soil microbiome based on 1512 soil metagenomic data. The topic is interesting as it helps us to understand the distribution and function of soil microbiomes and how they could change under scenarios of global change. However, the draft needs further improvement, and I have specific comments that need to be addressed.

Thank you for the positive comments!

1. Line 17, should specify the main objectives of the study, given that there are already patches of work addressing this topic.

Thank you for this suggestion. Given that this is the first sentence of the manuscript, we chose to leave this sentence broad in order to provide context for those unfamiliar with the field. We changed the sentence on lines 21-23 to more clearly state an objective. It now reads: "Our objective is to reveal novel biogeographical patterns of soil microbiomes across environmental factors and ecological biomes with high molecular resolution."

2. Line 19, why mention the release of greenhouse gases here?

We mention greenhouse gases because they are one of the critical functions of soil microorganisms that are alluded to in the previous sentence.

3. Lines 23-25 are hard to understand and require clarification.

We agree. We edited this sentence to include numbered phrases for more clarity. It now reads: "We reveal shifts in the potential for (1) microbial nutrient acquisition across pH gradients; (2) stress, transport, and redox-based processes across changes in soil bulk density; and (3) greenhouse gas emissions across biomes."

4. Line 50 should state that almost every aspect of the soil biogeochemical process is affected, not just "some."

While we agree with this reviewer, we believe that there may be some differences of opinion on the importance of microorganisms in all aspects of soil biogeochemistry. Abiotic processes like mineral weathering and hydrologic transport can also have very strong influences on soil biogeochemistry. Therefore, we chose to keep the word "some" to convey these nuances.

5. Line 67 should specify which factors matter.

Thank you. We adjusted the wording of this paragraph to improve clarity. This resulted in the deletion of the mention of factors here.

6. I strongly recommend that the authors focus more on the logical flow of the introduction, highlight the current knowledge gap, and explain how this study improves the knowledge of the readers rather than doing a literature synthesis.

We streamlined the introduction to enhance its readability and provide a clear objective statement.

7. Lines 105-106 require clarification.

We revised this sentence to read: “We collated soil metagenomic sequences from a wide range of environmental conditions across global biomes.”

8. After reading the results section, the main findings and the story that the authors want to express are not clear.

Thank you for this comment. We added stronger topic sentences to provide rationale and more structure to the results.

9. Line 206, the discussion section is too long, and I suggest condensing it and focusing on the main story and ecological implications.

Thank you. We tightened our writing in the discussion and provided more informative subheaders to increase readability.

10. Line 406 should address how to minimize any possible bias in meta-analysis, particularly regarding soil sampling and sequencing processes.

We added the following text to address this point (Lines 432-434): “While sample collection and sequencing methods inevitably vary in meta-analyses such as the current study, we applied standardized analytical workflows to minimize methodological artifacts to the greatest extent possible.”

11. Lines 429 and 447, results should be included, at least in the supplementary section.

We added Figure S1 and S2 to show these results.

12. Line 443, data were generated from global prediction and not observation data, raising concerns about the accuracy of the results.

While we agree that standardized measurements of all soil parameters is the gold standard, it is unfeasible for a study of this scale. There is a tradeoff between direct measurements and number of samples that are able to be included in meta-analyses. Our edaphic data are from the most comprehensive soil database available (ISRIC World Soil Information, <https://www.isric.org/>) and are inferred from a model with outstanding spatial coverage (~150,000 sites). Input data are shown in red points below (from Hengl et al. 2017). We also note that many of the sampling sites

included this meta-analysis (e.g., the National Ecological Observatory Network and the Earth Microbiome Project) also collect standardized soil properties for ISRIC.

13. Line 445 would benefit from a confusing matrix to understand the prediction power of this random forest model.

Confusion matrices are only applicable to categorical data. For random forest regressors, root mean square error is the most common metric for evaluating model performance. Only one of our variables was categorical (biome), while the rest were continuous. Therefore, to address this comment we added a supplemental table reporting the RMSE and R^2 of all continuous models (all variables except biome) and the out of bag error rate for classification models (biome, Table S1).

14. While the current results are mainly about the composition of soil microbiomes, metagenomic data mining should also include how functional genes and protein families distribute and change with environmental gradients.

We apologize for the confusion. We do not understand this comment since the majority of our results discuss functional genes (KO) and protein families (Pfam, TIGRFAM).

Reviewer #2 (Comments for the Author):

In this manuscript, the authors synthesize > 1500 metagenomes across continents and diverse ecosystems to determine the macroecology of soil microbial genomic potential and its ecological controls. Using both supervised and unsupervised random forest, the authors determine distinct genomic fingerprints of soil types and connect plausible mechanisms by which particular functional gene groups may relate to particular soil characteristics. Overall, this paper is sound and represents an impressive effort with a very large dataset to address a timely knowledge gap in microbial macroecology. I don't have any major critiques, but I think the manuscript would benefit from a dedicated discussion paragraph about the limitations of this effort/analysis.

Thank you for the positive comments!

For example, one limitation to discuss relates to In lines 435-436, where it seems like soil data like bulk density, cation exchange CEC, nitrogen, etc were not from the samples as sequenced by JGI but from GIS grids/pixels. I appreciate that temporally dynamic variables were not included, but some of these variables included can vary across a landscape, and this degree of variance may also differ by ecosystem type. For example, could the strength of associations between bulk density, pH, and genomic potentials and the lower strength of associations between genomic potentials with N and CEC be due to potentially greater landscape scale variance in N and CEC versus less landscape variability bulk density and pH?

We very much agree. In large meta-analyses such as this one, there is a tradeoff between breadth of data and precision of data. While directly measured standardized data are the gold standard, this would have significantly limited the number of samples available for this analysis. We note that data uncertainty should weaken observable relationships, and it is promising that we were able to detect meaningful relationships between the soil microbiome and some soil properties.

We added the following text to convey this on Lines 460-466:

“Additionally, because soil data are derived from a comprehensive spatial interpolation and not measured directly on each soil sample, our models may fail to capture local-scale heterogeneity in relationships. Further investigations that generate and leverage fully standardized data are needed to resolve this discrepancy. Nonetheless, local heterogeneity should statistically weaken the relationships observed here, and our results present a promising investigation into the global biogeography of soil microbiomes.”

Similarly, given conclusions about uncovering "microbial mechanisms underlying soil biogeochemistry", it would be helpful to discuss the limitations regarding the potential lack of coherence between functional gene abundance and biogeochemical function should be included.

We agree. Many works have failed to correlate microbial taxa and/or functional gene abundances with rates of their associated biogeochemical functions. Genes only convey the potential for a process rather than the rate itself and much of the downstream biology that may correlate more strongly with a rate is expensive and/or challenging to measure.

We added the following text to convey this on Lines 214-217:

“While we note that we assayed microbial genes which do not necessarily reflect expressed functions (61, 88–91), the genomic potential revealed by this investigation provides a basis for more targeted research into soil microbiome function across different soil habitats.”

Minor comments:

Line 255: Consider either expanding this paragraph or deleting it. It could be expanded by highlighting some examples of specific taxa at the genus or family level that are associated with vitamin B processes.

Thank you for this comment. Given the suggestion to shorten the discussion by reviewer 1, we significantly reduced this paragraph.

Line 274: are most of the high bulk density samples agricultural? If so, would be good to mention here.

We clarified this sentence to convey that land cover was not directly assessed (Lines 291-295). It now reads: “Though we did not directly assess land cover, the presence of potential functions associated with organic N cycling in the low bulk density fingerprint (K00992, K11959) may additionally suggest the influence of agriculture on soil microorganisms, where tillage generates loosely packed soils (i.e., low bulk density) and fertilizer application enhances nutrient cycles.”

Line 291: This sentence doesn't seem supported by data. Would be helpful to remind reader here about the result indicating associations between genes associated with methanogenesis and tundra samples.

We agree. Here, we are referring to known associations of methanogenic organisms with tundra and permafrost. We added citations and edited the language throughout this paragraph to make this clearer.

Re: mSystems01112-23R1 (Genomic fingerprints of the world's soil ecosystems)

Dear Dr. Emily B Graham:

Dear Emily, Thank you for the revised piece, which we are happy to accept. Congrats!

As a small note: I checked the FigShare for the manuscript code and noticed that there are several .DS_Store files that were possibly accidentally uploaded. Just letting you know so that these could be removed if needed.

Your manuscript has been accepted, and I am forwarding it to the ASM production staff for publication. Your paper will first be checked to make sure all elements meet the technical requirements. ASM staff will contact you if anything needs to be revised before copyediting and production can begin. Otherwise, you will be notified when your proofs are ready to be viewed.

Cover Image Submissions: If you would like to submit a potential Cover Image, please email a file and a short legend to msystems@asmusa.org. Please note that we can only consider images that (i) the authors created or own and (ii) have not been previously published. By submitting, you agree that the image can be used under the same terms as the published article. Image File requirements: TIF/EPS, 7.5 inches wide by 8.25 inches tall (at least 2,250 pixels wide by 2,475 pixels tall), minimum 300 dpi resolution (600 dpi preferred), RGB, and no figure elements, e.g., arrows or panel labels. The legend should be a short description of the image, 1-2 sentences recommended.

Sincerely,
Ashley Shade
Editor
mSystems